

**On the interconnections among major climate modes and their common**
**driving factors**
Xinnong Pan[1], Geli Wang[1*], Peicai Yang[1], Jun Wang[2], Anastasios A. Tsonis[3,4]
[1] Key Laboratory of Middle Atmosphere and Global Environment Observation (LAGEO), Institute of
Atmospheric Physics, Chinese Academy of Sciences, Beijing 100029, China
[2] Key Laboratory of Regional Climate-Environment for Temperate East Asia (RCE-TEA), Institute of
Atmospheric Physics, Chinese Academy of Sciences, Beijing 100029, China
[3] Department of Mathematical Sciences, Atmospheric Sciences Group, University of Wisconsin-Milwaukee,
Milwaukee 53201, USA
[4] Hydrologic Research Center, San Diego 92127, USA
*Correspondence to*: Geli Wang (wgl@mail.iap.ac.cn)





22          **Abstract**

The variations in oceanic and atmospheric modes on various timescales play important roles in generating
regional and global climate variability. Many efforts have been devoted to identify the relationships between
the variations in climate modes and regional climate variability, but rarely explored the interconnections
among these climate modes. Here we use climate indices to represent the variations in major climate modes
and we examine the harmonic relationship among the driving forces of climate modes by the combination of
Slow Feature Analysis (SFA) and wavelet analysis. We find that all of the significant peak-periods of
driving-force signals in the climate indices can be represented as the harmonics of four base periods: 2.32 yr,
3.90 yr, 6.55 yr and 11.02 yr. We infer that the period of 2.32 yr is associated with the signal of Quasi
Biennial Oscillation (QBO). The periods of 3.90 yr and 6.55 yr are connected with the intrinsic variability of
El Niño-Southern Oscillation (ENSO), and the period of 11.02 yr arises from the sunspot cycle. Results
suggest that the base periods and their harmonic oscillations linked to QBO, ENSO and solar activities act as
the key connections among the climatic modes with synchronous behaviors, highlighting the important roles
of these three oscillations in the variability of current climate.
**Key words:** climate modes; slow feature analysis; wavelet analysis; driving forces
**Highlights:**
i) The harmonic relationship among the driving forces of climate modes was investigated by using Slow
Feature Analysis and wavelet analysis.
ii) All of the significant peak-periods of driving-force signals in climate indices can be represented as the
harmonics of four base periods.
iii) The four base periods related to QBO, ENSO and solar activities act as the key linkages among different
climatic modes with synchronous behaviors.





## 1 Introduction


The influences of large-scale climate modes (e.g., El Niño-Southern Oscillation (ENSO), Pacific Decadal
Oscillation (PDO), North Atlantic Oscillation (NAO) and the Atlantic Multi-decadal Oscillation (AMO)) on
the variations of regional-to-global climate (e.g. temperature, rainfall, and atmospheric circulations) have
been extensively examined (Bradley et al., 1987; Wu et al., 2003; McCabe et al., 2004; Kenyon and Hegerl,
2008; Steinman et al., 2015; Wang et al., 2016; 2017; Yang et al., 2017; Zhang et al., 2017; Xie et al., 2019).
It has been well established that regional climate variations at various temporal and spatial scales are
modulated by the variabilities of major climate modes. For instance, Wu et al. (2003) estimated that about
25% rainfall variances in fall and winter over southern China can be explained by ENSO. McCabe et al.
(2004) reported that the PDO and AMO have contributed to more than half (52%) of the tempo-spatial
variance in multi-decadal drought occurrence over the conterminous Untied States. Xie et al. (2019) found
that the multi-decadal variability in East Asian surface air temperature (EASAT) is highly associated with
the NAO, which leads detrended annual EASAT by 15-20 years. Based on this relationship, they proposed a
NAO-based linear model to predict the near-future change in EASAT.

The variations of oceanic and atmospheric modes affect regional climate mainly through the teleconnections
within the atmosphere (i.e., atmospheric bridge) and ocean (i.e., oceanic tunnel) (Liu and Alexander, 2007).
Atmospheric teleconnections can be produced by both external forcings from ocean or land (e.g., sea surface
temperature (SST) anomalies related to ENSO) and internal atmospheric processes (e.g. Rossby wave in the
westerlies) (Trenberth et al., 1998). Though many theories have been developed to explain the physical
mechanisms behind the influences of major climate modes on regional climate, the interconnections among
these climate modes *per se*, and their primary driving factors remain largely unclear. Given that remote



teleconnections exist between climate modes and regional climate at various temporal and spatial scales,
tight interconnections are expected to exist among these climate modes (Rossi et al., 2011). In addition,
acted as the main energy source of the climate system, the external forcings of climate system (e.g., solar
activities) impose extensive influences on various climate modes (e.g., ENSO and NAO) (Kirov et al., 2002;
Velasco et al., 2008). Thus, it appears to be promising to identify the interconnections among major climate
modes and their common driving factors.

As the indicators of climate modes, many climate indices (e.g., the Niño3.4 SST for ENSO) have been
proposed and widely used to investigate the dynamic processes and physical mechanisms within climate
system (Dai, 2006; Steinman et al., 2015; Wang et al., 2017). However, the major barrier to clarify the
interconnections of these climate indices is how to effectively extract the driving forces, and identify their
corresponding essential driving factors. It is well recognized that most of the time series observed in the real
world are non-stationary because of the effects of external perturbations (Verdes et al., 2001). As a
representative non-stationary dynamic system, the driving forces for the variations of major climate modes
remain difficult to be determined. Some pioneering works have been conducted to solve this daunting
challenge. For example, Yang (2003) proposed a physical conceptual frame that the non-stationary features
of climate system are relevant to the characteristics of hierarchical structure: the driving force originating
from higher hierarchy sub-system controls the behaviors of lower hierarchy sub-system in a cascade way.
Compared to the dynamic reality as manifested in lower hierarchy sub-system, the driving force of higher
hierarchy sub-system tends to be a much slower process. In other words, the essential differences between
higher and lower sub-systems reflect in scale and energy.



Many efforts have been devoted to extract the information of driving force from dynamic system (Yang et
al., 2016). Slow feature analysis (SFA) is an algorithm that was developed to extract the slowly varying
features from non-stationary time series, which provides a direct and effective approach to identify the
driving forces of non-stationary dynamic system. Based on idealized models, recent studies have suggested
that the SFA can extract slowly-varying driving forces and sub-component signal from fast-varying
non-stationary time series even without any prerequisite knowledge about the underlying dynamic system
and its driving forces (Wiskott et al., 2002, 2003; Konen et al., 2009, 2011; Escalante-B et al., 2012).
Considering that the driving-force signal of dynamic system often consists of different components with
various time scales, Pan et al. (2017) detected the independent driving-force factors that contain significant
peak-periods from the SFA-extracted signals robustly through combing the SFA with wavelet analysis
(Torrence et al., 1998). Recently, this kind of technique that combines the SFA with wavelet analysis also
has been successfully applied to detect the external and internal driving-forces signals responsible for the
variations of regional climate, such as the drought variability in the southwestern United States (Zhang et al.,
2017), the temperature variations in the Central England (Wang et al., 2017) and the Northern Hemisphere
(Yang et al., 2016), and the oscillations of stratospheric ozone concentration (Wang et al., 2016). Thus, it is
reasonable to anticipate that this new approach can serve for the study of the interconnections among major
climate modes and their primary driving factors.

To this end, we aim to extract the driving forces of major climate modes and identify their interconnections
and primary driving factors by the combination of SFA and wavelet analysis. The remainder of this paper is
organized as follows. The data and methods used in this study are described in Sections 2 and 3, respectively.
The main results are presented in Section 4, followed by the conclusions and discussions in Section 5.



## 2 Data

In this study, we choose monthly mean indices for four widely-investigated climate modes (ENSO, PDO, AMO and NAO), which can be easily accessed from NOAA website (www…). Below, we will describe these four climate modes and their corresponding indices briefly.

### 2.1 ENSO

ENSO is well recognized as a natural ocean-atmosphere coupled mode in the tropical Pacific (Deser et al., 2010), which shows global impacts (Newman et al., 2003). El Niño (La Niña) refers to warming (cooling) phase of the tropical Pacific Ocean occurring every 2-7 yr. Meanwhile, the anomalous warming or cooling conditions are linked to a large-scale east-west seesaw air pressure pattern, referred to Southern Oscillation (Capotondi et al., 2015). El Niño and Southern Oscillation are two manifestations of ENSO phenomenon (Bjerknes, 1969). In this study, ENSO is represented by both Niño 3.4 SST anomalies and Southern Oscillation Indices (SOI). Niño 3.4 index (1870/01-2018/12, hereafter referred to as NINO) is defined as the SST anomalies in the Nino 3.4 region (5ºN-5ºS; 170-120ºW) based on the HadISST1 dataset (Rayner et al., 2003). SOI index (1866/01-2017/12) is calculated from the standardized sea level pressure (SLP) differences measured between the islands of Tahiti and Darwin, Australia (Ropelewski et al., 1987).

### 2.2 PDO

PDO is the dominant pattern of decadal variability of North Pacific SST, which has been widely-studied in various subjects (Newman et al., 2016). Previous study shows that the changing phase of PDO affects the anomalies of atmospheric circulation around North Pacific Ocean basin, and even the South Hemisphere (Mantua and Hare, 2002). The characteristic period of PDO is 50-60 yr and a warm or cold phase of PDO





can typically persist for about 20-30 yr. If PDO is in its positive phase, the North Pacific Ocean turns colder
and Middle East Pacific Ocean turns warmer, otherwise it is in negative phase. In this study, PDO is defined
by the leading principal component of monthly SST anomalies in the Pacific basin, poleward of 20ºN during
1900-2017 (Mantua et al., 1997).

**2.3 AMO**
AMO is a dominant signal of climate variability in the field of North Atlantic SST, which has a statistically
significant spectral peak in the 50-70 yr band (Schlesinger et al., 1994; Sun et al., 2015). Related studies
suggested that AMO is an inner variability of climate system, which can affect regional-to-hemispheric
climate (Zhang, 2007; Knight et al., 2006). The slow variation of the Atlantic meridional overturning
circulation (AMOC) is found to play a dominant role in the Atlantic multidecadal variability of SST (Zhang,
2017; Delworth et al., 2000; Garuba et al., 2018). In this study, AMO is defined as the detrended
area-weighted average SST over the North Atlantic from 0° to 70°N during 1856-2018 based on the Kaplan
SST dataset (Enfield et al., 2001). Two versions of AMO indices are available: unsmoothed one and
smoothed one. The high-frequency variability of the latter one has been removed out by a 121-month
smoother. We choose to use the unsmoothed AMO index in this study.

**2.4 NAO**
The NAO is active in the North Atlantic region that is characterized by a large-scale seesaw in atmospheric
mass between the subtropical high and the polar low (Li et al., 2003). It manifests as climate fluctuations at
multiple timescales ranging from inter-annual to multi-decadal variabilities (Jones et al., 1997; Li et al.,
2013), affecting the climate in and around North Atlantic Ocean basin, and even the entire Northern

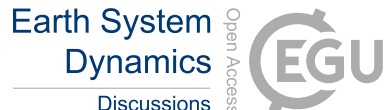

Hemisphere (Wallace and Gutzler, 1981; Hurrell,1995; Li et al., 2013; Delworth et al., 2016; Jajcay et al.,
2016;). Although NAO is most pronounced during the winter, it is the dominant mode of atmospheric
circulation in the North Atlantic sector throughout the whole year. Previous study suggested that NAO
drives the North Atlantic SST anomalies at the timescale less than 10 yr (Delworth et al., 2017).

NAO index is typically defined as a meridional dipole mode (which has been lately suggested of being a
three-pole pattern (Tsonis et al., 2008)) in atmospheric pressure with two centers of action in Iceland and
Azores during 1825-2017. For comparison, we also examine another observationally-based monthly NAO
index for the period of 1850-2015 (hereafter referred to as NAOI), which is defined as the difference in the
normalized sea level pressure (SLP) that is zonally-averaged over the North Atlantic sector from 80ºW to
30ºE between 35ºN and 65ºN (Li et al., 2003). NOAI is derived from the HadSLP dataset with the base
period of 1961-1990.

All the climate indices data mentioned above can be downloaded from the NOAA website of climate index
time series (www.esrl.noaa.gov/psd/gcos_wgsp/Timeseries/), except that NAOI is obtained from Prof.
Jianping Li's homepage in Beijing Normal University (http://ljp.gcess.cn/dct/page/65610). **Fig. 1** shows the
normalized native monthly time series of these climate indices.

**3 Methods**
**3.1 Slow Feature Analysis (SFA)**
Based on time-embedding theorems, one-dimensional time series can turn into a multidimensional system.
For this multidimensional input system, the SFA acts as a nonlinear method that uses a nonlinear expansion





to map the input signal into a feature space and solves a linear problem (Blaschke et al., 2006). The
objective of SFA is to find instantaneous scalar input-output functions that generate output signals that vary
as slowly as possible but still carry significant information. To ensure this, we require the output signals to
be uncorrelated and have unit variance (Franzius et al., 2011).

Consider a time series $\{x(t)\}_{t=t_1,\ldots,t_n}$, where $t$ stands for time and $n$ indicates the length of the time series.
First, we embed $\{x(t)\}$ into an $m$-dimensional state space:
$$\mathbf{X}(t) = \{x_1(t), x_2(t), \ldots, x_m(t)\}_{t=t_1,\ldots,t_N}$$
where $N = n - m + 1$. Then nonlinear expansions (usually second-order polynomials) are used to generate a
k-dimensional function state space:
$$\mathbf{H}(t) = \{x_1(t), \ldots, x_m(t), x_1^2(t), \ldots, x_1(t)x_m(t), \ldots, x_{m-1}^2(t), \ldots, x_m^2(t)\}_{t=t_1,\ldots,t_N},$$
which can also be written as $\mathbf{H}(t) = \{h_1(t), h_2(t), \ldots, h_k(t)\}_{t=t_1,\ldots,t_N}$, where
$k = m + m(m+1)/2$.

After that, the expanded signal $\mathbf{H}(t)$ is normalized so that it satisfies the constraints of zero mean and unit
variance. This process is referred to as whitening or sphering. Thus, we have
$$\mathbf{H}'(t) = \{h_1'(t), h_2'(t), \ldots, h_k'(t)\}_{t=t_1,\ldots,t_N}, \text{ where}$$
$$\overline{h_j'} = 0 \ \text{(zero mean)},$$
$$h_j' h_j'^T = 1 \ \text{(unit variance)},$$
$$h_j'(t) = [h_j(t) - \overline{h_j}]/S, \text{ and } S = \frac{1}{k}\sqrt{\sum_{j=1}^{k}(h_j(t) - \overline{h})^2}.$$
Then, by using Schmidz algorithm, $\mathbf{H}'(t)$ is orthogonized into:
$$\mathbf{Z}(t) = \{z_1(t), z_2(t), \ldots, z_k(t)\}_{t=t_1,\ldots,t_N}.$$



Thus, each output signal can be expressed as the following linear combination:
$$y(t) = a_1 z_1(t) + a_2 z_2(t) + \dots + a_k z_k(t),$$
$(a_1, a_2, \dots, a_k)$ is a set of weighting coefficients.

Note that the output signals are orthogonal and nontrivial:
$$z_i(t) \cdot z_j(t) = 0, \ \overline{z_i}(t) = \overline{z_j}(t) = 0, \ z_j(t) \cdot z_j^T(t) = 1,$$

Subsequently, we perform the 1ˢᵗ order differencing on $\mathbf{Z}(t)$ to obtain the derivative function space:
$$\dot{z}_j(t_i) = z_j(t_{i+1}) - z_j(t_i)$$
$$\dot{\mathbf{Z}}(t) = \{\dot{z}_1(t), \dot{z}_2(t), \dots, \dot{z}_k(t)\}_{t=t_1,\dots,t_N}.$$

Then we calculate the time-derivative $K \times K$ covariance matrix $\mathbf{B} = \dot{\mathbf{Z}}\dot{\mathbf{Z}}^T$, where its eigenvalues are $\lambda_1 \leq$
$\lambda_2 \leq \dots \leq \lambda_k$ and the corresponding eigenvectors are $\mathbf{W}_1, \dots, \mathbf{W}_k$. Finally, using $\mathbf{W}_1$, the driving force can
be written as:
$$y_1(t) = r\mathbf{W}_1 \cdot \mathbf{Z}(t) + c$$
where $r$ and $c$ are two arbitrary constants that resulting from quadrature of $y(t)$ and solution of $\mathbf{W}_1$,
respectively.

**3.2 Wavelet analysis**
Wavelet analysis is widely used to analyze localized structures and spectral properties of time series.
Torrence (1998) provided a useful toolkit to conduct wavelet analysis step by step including statistical



significance        testing.        The        toolkit        can        be        accessed        from        the        website:
http://paos.colorado.edu/research/wavelets/.

Here, we choose to use the Morlet wavelet, which offers a high spectrum resolution. The wavenumber is set
to 4, representing a lower resolution wavelet scale to analyze the time-averaged global power spectrum of
climate indices. Previous study based on idealized models shows that the significant peak-periods of the
SFA-derived signal correspond well to the driving force factors (Pan et al., 2017). Thus, we only focus on
the peak-periods that pass the significance test at the 95% confidence level in this study.

**4 Results**
As the first step, we set the embedding dimension $m$ to 11 (within one year) for the SFA and extract the
driving-force signals from climate indices, which are denoted as Snino, Ssoi, Spdo, Samo, Snao and Snaoi,
respectively. **Fig. 1** shows the variations of these SFA-extracted driving-force signals (red lines) along with
the native time series (grey lines) of climate indices. It should be noted that the slowly-varying signals
extracted by the SFA are essentially different from the low-frequency signal obtained by low-pass filtering.
In contrast to the quickly-varying and lack-of-feature native climate index time series, the slowly-varying
signals potentially represent the mixed effects of driving factors.

**Fig. 2** shows the time-averaged power spectrum of these driving-force signals as reconstructed by SFA. The
blue dots indicate the peak-periods that have passed the significant test at the 95% confidence level. Results
show that each SFA-extracted signal involves significant peak-periods at inter-annual to multi-decadal
timescales. **Table 1** lists the significant peak-periods of each climate indices**.** It is found that four base



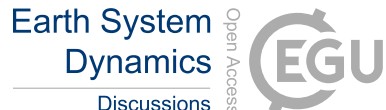

independent peak-periods (i.e. 2.32 yr, 3.90 yr, 6.55 yr and 11.02 yr) exist among different climate indices.
Other peak-periods of the SFA-derived signals from different climate indices can be expressed as integral
multiples of above base periods. For the sake of convenience, the above base peak-periods and their
corresponding harmonic periods are denoted by integral multiples of $T_q$ (purple), $T_{e1}$ (light blue), $T_{e2}$ (dark
blue) and $T_s$ (orange), respectively.

Furthermore, the peak-period of 2.32 yr ($T_q$, around 28 months) coincides with the cycle of quasi-biennial
oscillation (QBO) (Baldwin et al., 2001), which is the dominant pattern of variability in the tropical
stratosphere and displays alternating downward propagating easterly and westerly wind regimes. Although
the QBO is a tropical stratospheric phenomenon, it not only affects the chemical constituents (e.g. water
vapor, and ozone etc.) but also affects the stratospheric flow from pole to pole by changing the influences of
extra tropical waves. Specifically, through the effects on polar vortex, QBO can affect the surface weather
patterns indirectly (Baldwin et al., 2001). Previous studies suggested that the temperature gradient between
the troposphere and stratosphere can modulate the Walker circulation and SST anomalies in equatorial
Pacific Ocean by altering the atmospheric stability and tropical deep convection (Huang et al., 2011).

We cautiously infer that the two periods (i.e. 3.90 yr ($T_{e1}$) and 6.55 yr ($T_{e2}$)) are related to the intrinsic
inter-annual variability of ENSO activities, and the period of 11.02 yr ($T_s$) corresponds well to the Schwabe
sunspot cycle (11 yr). The results of harmonic analysis shows that the peak-periods of the SFA-derived
signals from different climate indices can be expressed as integral multiples of base independent periods (i.e.
$T_q$, $T_{e1}$, $T_{e2}$ and $T_s$), implying that these four independent periods associated with QBO, ENSO and solar





activity can be regarded as three common driving factors for the variabilities of ENSO, PDO, AMO and
NAO.

Given that the driving-force signal consists of several components, the selection of embedding dimension $m$
might affect the result of phase-space reconstruction of time series (Konen et al., 2009; Yang et al., 2016).
Considering that the peak-periods of SFA-extracted driving-force signals may be sensitive to the embedding
dimension $m$ as set in SFA, we conduct additional analysis by increasing $m$ from 1 to 25 months (covering
two years) to detect the significant peak-periods of these driving-force signals. As **Fig. 3** shows, all the
significant peak-periods can be represented as the integral multiples of $T_q$, $T_{e1}$, $T_{e2}$ and $T_s$, which confirms
that above-mentioned three driving factors (QBO, the intrinsic variabilities of ENSO, and solar activities)
are the common driving factors for the variabilities of ENSO, PDO, AMO and NAO. Note that the
significant peak values on a longer time scale are sensitive to the setting of high embedding dimensions,
which is probably associated with the smoother SFA signals when setting higher embedding dimensions.

We further exploit the information involved in **Fig. 3** and decompose them into following tables. **Table 2**
shows the number of embedding dimensions by which a peak period is significant for each index. The two
columns show periods and the corresponding identifier (forcing), respectively. If this number is greater than
10, it is highlighted in bold. Taking Snino for example, the entries in **Table 2** show that 15/25 embedding
dimensions have significant peak-value at the period of 74.13 yr ($32T_q$); 12/25 embedding dimensions have
significant peak-value at the period of 3.90 yr ($T_{e1}$); 16/25 embedding dimensions have significant
peak-value at the period of 5.51 yr ($0.5T_s$); and 17/25 embedding dimensions have significant peak value at
the period of 11.02 yr ($T_s$).





As shown in **Table 2**, each climate mode can be modulated by various driving factors that generate
harmonic oscillations at different timescales. For instance, QBO presents four harmonic oscillations from
inter-annual (9.27 yr) to multi-decadal (74.13 yr) periods on NINO variability; The intrinsic variability of
ENSO presents five harmonic oscillations from intra-seasonal (0.2 yr) to multi-decadal (52.42 yr) timescales
on the NAO variability. Similar conditions can be found for other climate indices.

In addition, it is found that different climate indices involve same driving harmonic oscillations. For instance,
both PDO and AMO are modulated by the period of 9.27 yr, which is a QBO-related harmonic oscillation;
both NINO and SOI are modulated by the period of 3.90 yr, which we infer is related to intrinsic ENSO
cycle; both NINO and PDO are modulated by the inter-annual period of 5.51 yr, which is a harmonic
oscillation of solar activity.

The results displayed in **Fig. 3** and **Table2** can be alternatively presented in **Tables 3** and **4**. In **Table 3**, the
columns are the driving force factors (Tq, Te1, Te2 and Ts) and the rows are the climate indices. The entries
in the table show the harmonic(s) of driving force factors affecting each index in more than 10 embedding
dimensions. It shows that ENSO-Te1 presents the least number of harmonic peak-periods, and that solar,
QBO and ENSO-Te2 present equally similar number of peak-periods in shaping the variability of climate
indices. Finally, **Table 4** shows the corresponding driving harmonic oscillations that modulate the variability
of climate indices in various time scales (periods) for all embedding dimensions. The entries in bold
correspond to the gray entries in **Table 2**.
In most conditions, as shown in **Table 4**, the driving harmonic oscillations among different climate indices
are diverse and complicated in the periods less than 20 yr. Take NAOI for example, there could be up to five





driving harmonic oscillations in similar time scales (1-5 yr). Nevertheless, driving harmonic oscillations in
the multi-decadal period of 50-55 yr are only related to ENSO –Te2, and the ones in the period of 60-65 yr
are only associated with ENSO –Te1. For the driving harmonic oscillations in the period of 70-75 yr, the
QBO is identified as the primary influencing factor. The driving harmonic oscillations in the period of 80-85
yr are found to be linked to Ts. Above results are useful for improving our understanding of climate
variability in different time scales.

Based on the results obtained by combining SFA with wavelet analysis, we find that all the detected
peak-periods can be represented as the integral multiples of the base peak-periods associated with QBO,
intrinsic variabilities of ENSO and solar activities. Considering that the time series of AMO used in this
study is unsmoothed, we repeat the analysis by using the smoothed AMO index (with a 121-month
smoother). The peak-periods detected in the smoothed time series are exactly the same with the ones based
on unsmoothed index (figure not shown). This suggests that the pre-processing of the AMO index has little
influence on the application results of SFA.

**5 Conclusions and discussions**
In this study, we identify four independent base peak-periods: $T_q$ (2.32 yr), $T_{e1}$ (3.90 yr), $T_{e2}$ (6.55 yr) and $T_s$
(11.02 yr). We infer that these four base peak-periods are essentially associated with the QBO cycle, two
intrinsic ENSO cycles and the solar cycle, respectively. Other detected significant peak-periods can be
represented by the integral multiples of these four base periods. It implies that the QBO, ENSO and solar
activities could be three key periodic driving factors in global climate variability. These results provide
possible clues for the intricate relationships between driving forces and their harmonics in the variability of

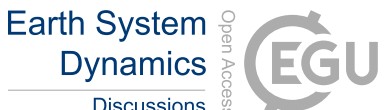

major climate modes as well as their corresponding coupling ways. The existence of these interconnections
of major climate modes indicates that it is promising to develop statistical models to predict the
decadal-to-multidecadal climate variability. It should be noted that uncertainties still remain in the
multidecadal variability of ENSO and QBO. The relatively long peak-periods (e.g., 52.42 yr, 62.33 yr, 74.13
yr and 88.15 yr) detected by SFA may be caused by the influence of the continuous wavelet transform.

Recent studies on complex climate networks provided new insights into how the collective behavior of
major climate modes affects global temperature variations (Tsonis et al., 2007; Tsonis 2018). By considering
a network of major climate modes (more or less the same set as here) and the theory of synchronized chaos,
these previous studies found that the network may synchronize temporally. During synchronization, the
increased coupling strength among the climate modes may lead to the destruction of the synchronized state
that leads to changes in the trends of global temperature and the amplitudes of ENSO variability in
decadal-to-multidecadal timescales. These studies proposed a dynamical mechanism and its related physical
causes for the observed climate shifts. Our results provide further new insights into those physical
mechanisms and how the complex interactions among the base driving factors and their harmonics cause the
peak-periods in climate modes.

***Code/Data availability.*** All data needed to evaluate the conclusions in the paper are present in the paper.
Additional data and codes related to this paper may be requested from the corresponding author.

***Author contribution.*** Xinnong Pan and Geli Wang designed this study. All of the authors contributed to
the preparation and writing of the manuscript.




***Competing interests.*** The authors declare no competing interest.

***Acknowledgements.*** This research was supported by the National Key R&D Program of China
(2017YFC1501804), the National Natural Science Foundation of China (91737102 and 41575058).

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





**Table 1**. The peak-periods of SFA-extracted slow feature signals and their classification.

| Snino | 3.90 | 5.51 | 11.02 | 74.13 | | | |
|---|---|---|---|---|---|---|---|
| | $T_{e1}$ | Ts/2 | Ts | 32Tq | | | |
| Ssoi | 3.90 | 6.55 | 11.02 | 52.42 | | | |
| | $T_{e1}$ | Te2 | Ts | 8Te2 | | | |
| Spdo | 5.51 | 9.27 | 18.53 | 37.06 | 62.33 | | |
| | Ts/2 | 4Tq | 8Tq | 16Tq | 16Te1 | | |
| Samo | 9.27 | 26.21 | 52.42 | 74.13 | | | |
| | 4Tq | 4Te2 | 8Te2 | 32Tq | | | |
| Snao | 2.75 | 7.79 | 11.02 | 62.33 | 88.15 | | |
| | Ts/4 | 2Te1 | Ts | 16Te1 | 8Ts | | |
| Snaoi | 2.75 | 4.63 | 7.79 | 13.10 | 26.21 | 44.08 | 88.15 |
| | Ts/4 | 2Tq | 2Te1 | 2Te2 | 4Te2 | 4Ts | 8Ts |




















**Table 2**. The entries show for each index, the number of embedding dimensions in which a peak period is
significant. The left column lists the period and the right column the identifier (forcing). If this number is
greater than 10 is highlighted in bold.

| Periods | Snino | Ssoi | Spdo | Samo | Snao | Snaoi | identifier |
|---|---|---|---|---|---|---|---|
| 0.58 | | | | | | 1 | $0.25T_q$ |
| 1.16 | | | | | 4 | | $0.5T_q$ |
| 2.32 | | | | | | 4 | $T_q$ |
| 4.63 | | | | | | 7 | $2T_q$ |
| 9.27 | 1 | | **25** | **25** | | | $4T_q$ |
| 18.53 | 5 | | **15** | | **14** | | $8T_q$ |
| 37.06 | 7 | | **24** | | | | $16T_q$ |
| 74.13 | **15** | 7 | | **25** | | | $32T_q$ |
| 0.49 | | | | | 2 | | $T_{e1}/8$ |
| 0.97 | | | | | 3 | 6 | $T_{e1}/4$ |
| 3.90 | **12** | **13** | | | | 6 | $T_{e1}$ |
| 7.79 | 7 | | | | 9 | 9 | $2T_{e1}$ |
| 15.58 | 5 | | | | | 1 | $4T_{e1}$ |
| 62.33 | | | **25** | | **17** | | $16T_{e1}$ |
| 0.20 | | | | | | 1 | $T_{e2}/32$ |
| 3.28 | | 6 | | | **10** | 3 | $0.5T_{e2}$ |
| 6.55 | | **20** | 6 | | | | $T_{e2}$ |
| 13.10 | 3 | **12** | 7 | | | 3 | $2T_{e2}$ |
| 26.21 | | 7 | | **25** | | 4 | $4T_{e2}$ |
| 52.42 | | **17** | | **25** | | **11** | $8T_{e2}$ |
| 2.75 | | | | | 5 | 4 | $0.25T_s$ |
| 5.51 | **16** | | **19** | | | | $0.5T_s$ |
| 11.02 | **17** | **11** | | | **20** | | $T_s$ |
| 22.04 | | | | | | **12** | $2T_s$ |
| 44.08 | | | | | | **10** | $4T_s$ |
| 88.15 | | | | | **15** | **23** | $8T_s$ |














**Table 3**. The entries in the table show the harmonics of the basic driving forces (significant when affecting an index in more than 10 different embedding dimensions) for each climate mode index.

| Climate Indices | $T_q$ (QBO) | $T_{e1}$ (ENSO) | $T_{e2}$ (ENSO) | $T_s$ (solar) |
|---|---|---|---|---|
| Nino | 32 | 1 | - | 0.5, 1 |
| SOI | - | 1 | 1, 2, 8 | 1 |
| PDO | 4, 8, 16 | 16 | - | 0.5 |
| AMO | 4, 32 | - | 4, 8 | - |
| NAO | 8 | 16 | 0.5 | 1, 8 |
| NAOI | - | - | 8 | 2, 4, 8 |

**Table 4**. The basic driving forces and their harmonic oscillations that are associated with the variability of climate mode indices at various time scales (periods) for all embedding dimensions. The entries in bold correspond to the highlighted numbers in **Table 2**.

| Scales | Snino | Ssoi | Spdo | Samo | Snao | Snaoi |
|---|---|---|---|---|---|---|
| <1y | | | | | $T_{e1}/8, T_{e1}/4$ | $0.25T_q, T_{e1}/4, T_{e2}/32$ |
| 1-5y | $\mathbf{T_{e1}}$ | $\mathbf{T_{e1}}, 0.5T_{e2}$ | | | $0.5T_q, \mathbf{0.5T_{e2}}, 0.25T_s$ | $T_q, 2T_q, T_{e1}, 0.5T_{e2}, 0.25T_s$ |
| 5-10y | $4T_q, 2T_{e1}, \mathbf{0.5T_s}$ | $\mathbf{T_{e2}}$ | $\mathbf{4T_q}, T_{e2}, \mathbf{0.5T_s}$ | $\mathbf{4T_q}$ | $2T_{e1}$ | $2T_{e1}$ |
| 10-15y | $2T_{e2}, \mathbf{T_s},$ | $\mathbf{2T_{e2}, T_s}$ | $2T_{e2}$ | | $\mathbf{T_s}$ | $2T_{e2}$ |
| 15-20y | $8T_q, 4T_{e1}$ | | $\mathbf{8T_q}$ | | $\mathbf{8T_q}$ | $4T_{e1}$ |
| 20-25y | | | | | | $\mathbf{2T_s}$ |
| 25-30y | | $4T_{e2}$ | | $\mathbf{4T_{e2}}$ | | $4T_{e2}$ |
| 30-35y | | | | | | |
| 35-40y | $16T_q$ | | $\mathbf{16T_q}$ | | | |
| 40-45y | | | | | | $\mathbf{4T_s}$ |
| 45-50y | | | | | | |
| 50-55y | | $\mathbf{8T_{e2}}$ | | $\mathbf{8T_{e2}}$ | | $\mathbf{8T_{e2}}$ |
| 55-60y | | | | | | |
| 60-65y | | | $\mathbf{16T_{e1}}$ | | $16T_{e1}$ | |
| 65-70y | | | | | | |
| 70-75y | $\mathbf{32T_q}$ | $32T_q$ | | $\mathbf{32T_q}$ | | |
| 75-80y | | | | | | |
| 80-85y | | | | | | |
| 85-90y | | | | | $\mathbf{8T_s}$ | $\mathbf{8T_s}$ |


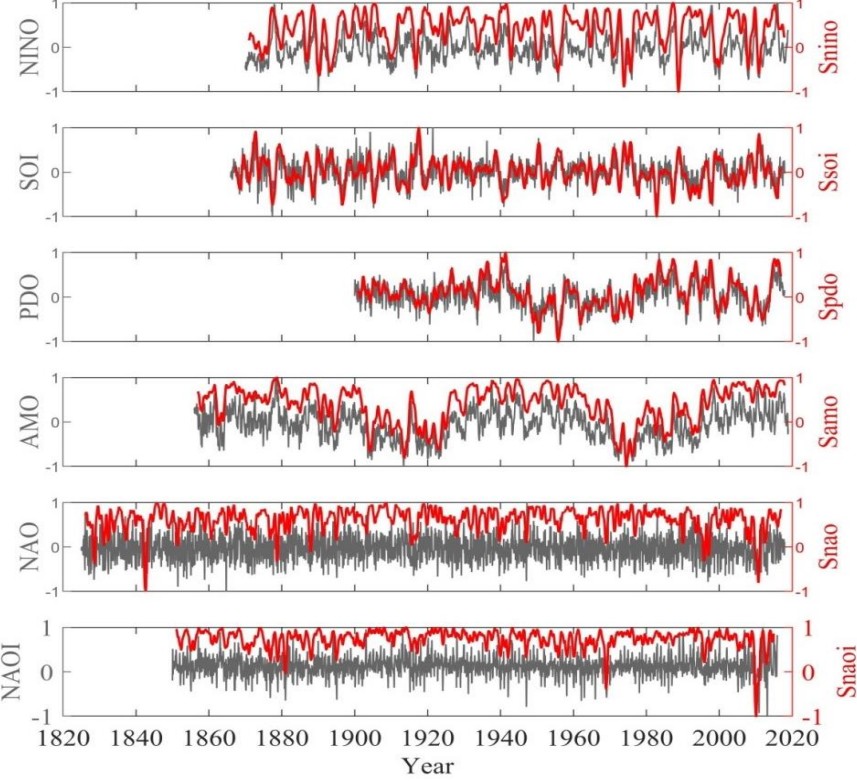


**Figure 1**. Normalized monthly time series of six climate indices during each periods (gray lines): NINO
(01/1870–12/2018), SOI (01/1866–12/2017), PDO (01/1900–12/2017), AMO (01/1856–12/2018), NAO
(01/1825–12/2017) and NAOI (01/1850–12/2015); And their corresponding SFA-derived slow feature
signals (red lines) , which are indicated by Snino, Ssoi, Spdo, Samo, Snao and Snaoi, respectively (setting
embedding dimension m to be 11).

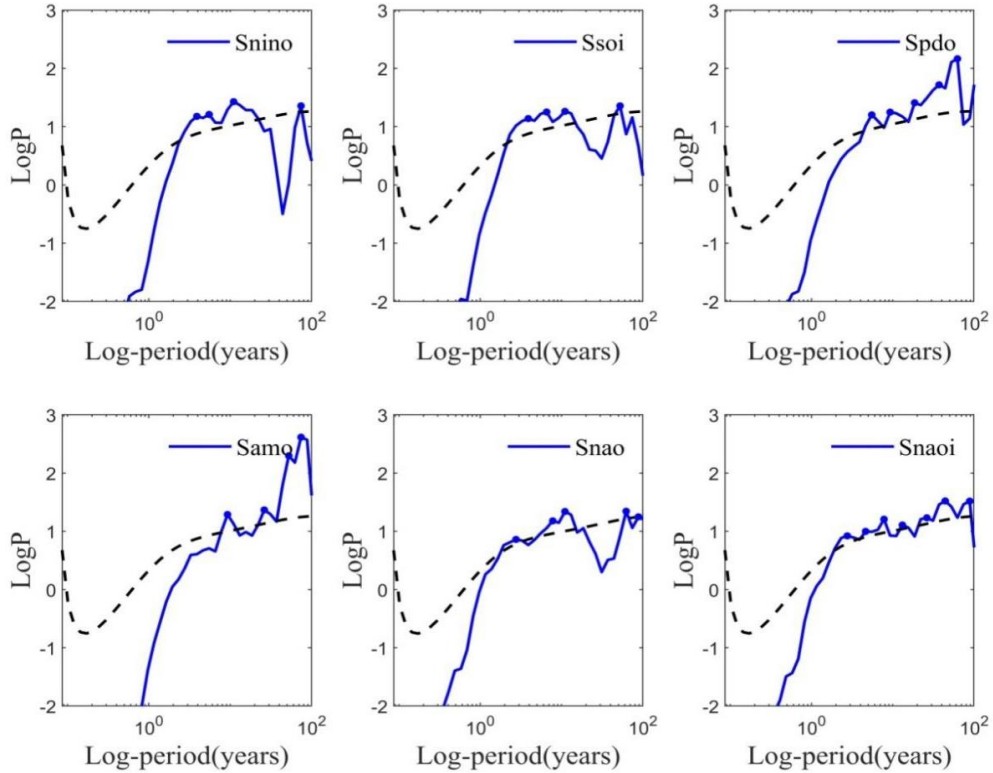


**Figure 2**. The time-averaged power spectrum of SFA-extracted (m=11) slow feature signals for six climate
indices, and the significant points (blue dots) with peak power that pass the significance test at the 95%
confidence level (black dashed lines) are also indicated.



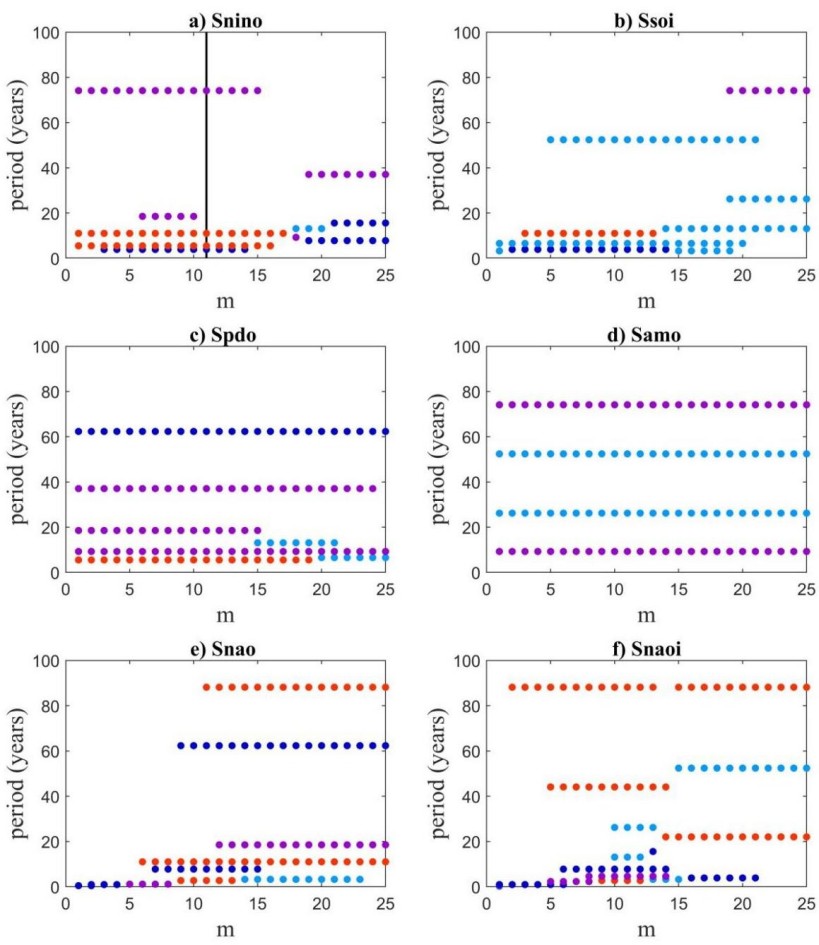


**Figure 3**. The significant peak periods of the SFA-extracted slow feature signals in six climate indices when

setting different embedding dimensions from 1 to 25.