# Peer review of "On the interconnections among major climate modes and their common driving factors"

_Earth System Dynamics, 2019_

## Short Comment (SC1) · 3 Jan 2020

If our final research objective consists in the long-term weather prediction and the short-term outlook of the climate change then it is important to know and understand processes in the climate system of larger spatial and temporal scales than those inherent to the cyclonic activities. It is well known that Sir Walker and his colleagues (Berlage and some others) were the pioneers in such studies (detecting the Southern Oscillation in tropics). Much later, Wallace and others discovered the so-called teleconnections in extratropics. After this, it became clear that there is a certain internal order in seemingly chaotic climate dynamics. Now it is the generally accepted to believe that ENSO is the main component of this order influencing climatic processes almost everywhere on the Earth. But there are also (not numerous) indications that some of extratropical

teleconnections themselves affect ENSO. Therefore, the question of what is the root cause of the multiscale climate dynamics remains to be open. By this reason, the topic of the paper being discussed seems to be appropriate. However, it is necessary to remember that, according to the second law of the thermodynamics, no order occurs in a dynamical system if there are no external factors affecting this system. In connection with this, the conclusion of the paper being discussed that effects of all individual climatic processes on each other can be reduced to the influence of only two of them: the quasibiennial oscillation (QBO) in the lower stratosphere and ENSO. Of course, a reference is done to the solar activity which is one of the external factors possible affecting the climate system. It is because the paper authors found four main peaks in the power spectra: of a set of the tropical and extratropical teleconnection indices: 2.32, 3.90, 6.55, and 11.02 years. In fact, a different interpretation of these peaks can be given. The period of 2.32 years (28 months) is exactly equal to the doubled main period of the so-called Chandler wobble in the Earth's pole motion. It seems to be appropriate to consider the Chandler wobble as the prime cause of QBO, and so an indirect cause of other teleconnections. The periods of 3.90 and 6.55 are not the main periods of ENSO. According to numerous calculations, the most pronounced period of ENSO is 3.5 years (42 months). It is exactly the trebled (the subharmonic 3:1) Chandler wobble period. Besides, there exist several other ENSO peaks in the range between 2 to 7 years. Some of which are equal to other subharmonics (e.g. 2:1, 4:1) of the Chandler wobble period, i.e. $\sim$2.4 and $\sim$4.8 years. The period of $\sim$6.2 years (instead of 6.5 of the authors) seems to be equal to the superharmonic 1:3 of the Luni-Solar nutation (the main period = $\sim$18.6 years). Thus, the Luni-Solar nutation seems to be another prime cause of ENSO and others. Thus, both afore-mentioned external factors must be indicated in the paper discussed as the prime causes of the macroscale climatic processes instead of QBO and ENSO. One can mention that the most recent and careful computation of the ENSO power spectra is published in Serykh I.V., Sonechkin D.M. "Nonchaotic and globally synchronized short-term climatic variations and their origin". Theor. Appl. Climatol., 2019, DOI 10.1007/s00704-018-02761-0. Moreover, in this paper pactically the same set of power energy peaks is found in the power spectra of other teleconnections which form (together with ENSO) a global scale spatial structure called the Global Atmospheric Oscillation (GAO) in Serykh I.V. et al. "Global Atmospheric Oscillation: an integrity of ENSO and extratropical teleconnections". PAGEOPHYS, 2019, https://doi.org/10.1007/s00024-019-02182-8.

---

## Author Comment (AC1) · 5 Jan 2020

We thank Prof. Sonechkin for his comments. We are willing to discuss his points on the Chandler wobble and the Luni-Solar nutation and include his references in the revised/final version of the paper.
* * *

---

## Referee Comment (RC1) · Anonymous Referee #1 · 6 Jan 2020

This is an interesting work to explore the interconnections among different climate modes. They found there are four basic periods in different climate indices, whileas the other periods can be taken as the homonics of these four basic periods.

There are some issues should be well addressed before the acceptance os this submission,

First of all, for the same climate mode, such as NAO, different indices reach different period results (see Fig. 2), why? The authors should provide their explanations to these differences.

Secondly, for the common basic perdiods to all studied climate indices, the physical implications should be provided.

[Figure]

Thirdly, The peak values for each period are given, how about their uncertainties?
* * *

---

## Referee Comment (RC2) · Anonymous Referee #2 · 18 Jan 2020

Review of MS No.: esd-2019-74

Title: On the interconnections amongmajor climate modes and their common driving factors

Author(s): Xinnong Pan, Geli Wang, Peicai Yang, Jun Wang, and Anastasios A. Tsonis

The authors apply the Slow Feature Analysis (SFA) on climate indices representing the variations in major climate modes in order to extract driving forces and, applying the wavelet analysis, they estimate the main oscillatory components characterizing the variability of the analyzed climate modes. About 26 different periods are interpreted as harmonics of only four main periods rooted in QBO, ENSO and the solar activity. This seems as a very interesting and important result, however, I am afraid that

such a bold conclusion requires more than just estimating the periods. Moreover, as already the first referee pointed out, the authors should present the mean estimates of the periods and their errors in order to see how well particular periods fit into the harmonic relationships. It would be also interesting which physical mechanisms would lead to occurrences of 16- or 32fold multiples of the basic periods. The authors write about synchronous behavior or harmonic relationships, however, no relationships between the identified modes are tested. The simple list of estimated periods is not sufficient for such conclusions. The wavelet analysis gives the possibility to extract the oscillatory components themselves, so relations of them should be investigated. For instance, if a mode contains an oscillatory component which is a harmonic of a basic period of another mode, than one would expect some coherence, or phase synchrony of these components obtained from different climate modes. There are also tools for identification of higher harmonics in experimental data, see, e.g. Sheppard, L. W., A. Stefanovska, and P. V. E. McClintock. "Detecting the harmonics of oscillations with time-variable frequencies." Physical Review E 83.1 (2011): 016206.

---

## Author Comment (AC2) · 20 Feb 2020

**(1) Comments from Referee #1**

This is an interesting work to explore the interconnections among different climate modes. They found there are four basic periods in different climate indices, whereas the other periods can be taken as the harmonics of these four basic periods. There are some issues should be well addressed before the acceptance of this submission. First of all, for the same climate mode, such as NAO, different indices reach different period results (see Fig. 2), why? The authors should provide their explanations to these differences. Secondly, for the common basic periods to all studied climate indices, the physical implications should be provided. Thirdly, the peak values for each period are given, how about their uncertainties?

**(2) Author's response**

Thank you for your encouraging comments.

We sincerely appreciate this helpful and insightful comments and the corrections to English that you have provided. Basically, there are three comments: 1) why the difference between NAO and NAOI in fig. 2; 2) what are the physics here, and 3) what are the uncertainties in the peak values. Comments 1 and 2 are related and we will discuss them together.

Comments 1 and 3

**The definitions of two NAO indices we used in this study are different.** One is defined as the difference in monthly atmospheric pressures between two action centers in Iceland and Azores during 1825-2017. Another observationally-based monthly NAO index for the period of 1850-2015 (referred to as NAOI) is defined as the difference in the normalized sea level pressures (SLP) that are zonally-averaged over the North Atlantic sector from 80ºW to 30ºE between 35ºN and 65ºN. NOAI is derived from the HadSLP dataset with the base period of 1961-1990.

Figure 2, is an illustration for an embedding dimension to 13 to detect the peak-periods of the driving factors of NAO (SFA-NAO) and NAOI (SFA-NAOI). These results suggest that the peak-periods of SFA-NAO are related to ENSO and solar activities, and the peak periods of SFA-NAOI are associated with ENSO, QBO and solar activities (see Figure 2 and Figure 3). However, as Figure 3 shows, when we vary the embedding dimensions from 1 to 25, the peak-periods of both SFA-NAO and SFA-NAOI show robust relations with ENSO, QBO and solar activities. In a way, repeating for many embedding dimensions serves as a sensitivity analysis to see if the results are robust. To further demonstrate this we present some arguments based on a simple known system.

In this study, we did not quantify the uncertainties of the peak values. Here, we use an ideal model to show the change of the significant peak periods of SFA-derived signals among different embedding dimensions. Results show that the peak periods of SFA signal are enough to represent the characteristics of the driving force factors, either the combination of the independent driving force factors or only the slowest driving force factor. Sensitive test of SFA with different embedding dimensions is an effective way to exacting the information of driving force factors.

Consider a logistic map with two-level structures which contains two driving time-varying factors a($t$) and b($t$):

$$x(t + 1) = a(t)x(t)(b(t) - x(t)), (t = 1,2,\ldots,n) \quad \ldots(1)$$

Where

$$a(t) = 3.5 + 0.4\cos(2pt/78) \quad \ldots(2)$$

$$b(t) = 0.998 + 0.024\cos(2p^2t/78) \quad \ldots(3)$$

From formulas (2) and (3), we can compute the true periods of a($t$) and b($t$) are nearly 78.00 and 24.84 steps, respectively.

First, we set the embedding dimension $m$ to 6 for the SFA and extract the driving-force signal from x($t$), which is denoted as sfa-x($t$).

Figure 1s a and 1s b below show the time series of the parameters (a($t$) and b($t$)) in the Logistic map with two-level structure. Figure 1s c shows the non-stationary time series x($t$), which is controlled by a($t$) and b($t$). The driving signal of x(t) when setting embedding dimension m to be 6 (i.e. sfa-x($t$)) is shown in Figure 1s d. From Figure 1s, we can see that the temporal evolution of sfa-x($t$) is neither a($t$) or b($t$), but the combination form of them. As a result, we can detect the driving information of sfa-x($t$) robustly through wavelet analysis. Figure 2s shows the time-averaged power spectrum of each parameter in Figure 1s. We can see that the significant peak periods in Figure 2s a and Figure 2s b are very close to the true period of a($t$) and b($t$). The significant peak periods of sfa-x($t$) are equal to the significant peak periods of a($t$) and b($t$). It means that the significant peak periods of SFA-derived signals represent the periodic characteristics of detected driving forces very well. As we mentioned above, only one embedding dimension is not enough to demonstrate the effectiveness of SFA. To reconfirm the results derived from SFA and wavelet analysis, we carry out a series of sensitivity tests to identify the significant peak periods of SFA-derived signals by setting the embedding dimensions from 2 to 16. The peak periods in the time-averaged power spectrum of SFA signals (significant at a significance level of 0.05) with different embedding dimensions are shown in Figure 3s. Results show that under different embedding dimension, SFA signal is ether the combination of different independent driving force factor, or

only the slowest driving force factor (higher embedding dimensions tend to smooth out the faster oscillations).

[Figure]

Figure 1s The time series of the parameters in the Logistic map with two-level structure. (a) Time series of true driving forces $a(t)$; (b)Time series of true driving force $b(t)$; (c) The non-stationary time series $x(t)$, which is controlled by $a(t)$and $b(t)$; (d) The slow feature $sfa\text{-}x(t)$ derived from $x(t)$by using SFA ($m = 6$and $t = 1$).

[Figure]

Figure 2s From (a) to (d) are the time-averaged power spectrums (black lines) of $a(t)$, $b(t)$, $x(t)$ and sfa-x(t), respectively. The red dashed lines show the significant test at the significance level of 0.05 for wavelet analysis. The red dots indicate the peak periods that can pass the significant test. The values of corresponding period are denoted along with red dots.

| m | Periods
(reserve two decimal fractions) | |
|---|---|---|
| 2 | **23.38** | **78.62** |
| 3 | **23.38** | **78.62** |
| 4 | **23.38** | **78.62** |
| 5 | **23.38** | **78.62** |
| 6 | **23.38** | **78.62** |
| 7 | **23.38** | **78.62** |
| 8 | **23.38** | **78.62** |
| 9 | **23.38** | **78.62** |
| 10 | **78.62** | |
| 11 | **78.62** | |
| 12 | **78.62** | |
| 13 | **78.62** | |
| 14 | **78.62** | |
| 15 | **78.62** | |
| 16 | **78.62** | |

Figure 3s The peak periods in the time-averaged power spectrum of SFA signals (significant at a significance level of 0.05) when setting different embedding dimensions (unit: step).

To sum up, based on a series of sensitivity test with an ideal model, we demonstrate the effectiveness and robustness of the technique of combining SFA with wavelet analysis. The sensitivity analysis of varying the embedding dimension appear so provide robust results: **The significant peak-periods of SFA signals can reflect the true driving forces very well. We note that other methods may be developed to quantify the uncertainties of the peak periods in future studies.**

**Comment 2**

Here are some additional facts that we are planning to add in order to connect our results to known dynamical mechanisms in decadal to multi-decadal variability. In this study, we identify four base periods in the driving signals of four climate modes as 2.32 yr, 3.90 yr, 6.55 yr and 11.02 yr, which are inferred to be associated with the signals of QBO, ENSO and sunspot cycle. Though they exhibit different coherent periods in the driving signals of different climate indices, our results provide some clues for the intricate relationships between driving forces and their harmonics in the variability of major climate modes as well as their coupling ways.

Recent studies on complex climate networks provided new insights into how the collective behavior of major climate modes affects global temperature variations (Tsonis et al., 2007; Tsonis 2018). By considering a network of major climate modes (more or less the same set as here and the theory of synchronized chaos, these previous studies found that the network may synchronize temporally. During synchronization, the increased coupling strength among the climate modes may lead to the destruction of the synchronized state that leads to changes in the trends of global temperature and the amplitudes of ENSO variability in decadal-to-multidecadal timescales. These studies proposed a dynamical mechanism and its related physical causes for the observed climate shifts. The idea that the interaction between major climate modes play a significant role in climate variability has in the last decade or so found many applications.

Solid dynamical arguments and past work offer a concrete picture of how the physics may play out (see G. Wang, K.L. Swanson, and A.A. Tsonis, 2009: The pacemaker of major climate shifts. Geophys. Res. Lett., doi:10.1029/2008GL03684 and references therein). NAO with its huge mass re-arrangement in north Atlantic affects the strength of the westerly flow across mid-latitudes. At the same time through its "twin", the arctic Oscillation (AO), it impacts sea level pressure patterns in the northern Pacific. This process is part of the so-called intrinsic mid-latitude northern hemisphere variability. Then this intrinsic variability through the seasonal "footprinting" mechanism couples with equatorial wind stress anomalies, thereby acting as a stochastic forcing of ENSO. Subsequently, ENSO with its effects on PNA can through vertical propagation of Rossby waves influence the lower stratosphere and in turn the stratosphere influence NAO through downward progression of Rossby waves. These results coupled with our results suggest the following 3-D super-loop NAO → PDO → ENSO → PNA → stratosphere → NAO, which may capture the essence of low-frequency variability in the northern hemisphere (Figure 4s).

[Figure]

Figure 4s 3-D super-loop NAO → PDO → ENSO → PNA → stratosphere → NAO, which may capture the essence of low-frequency variability in the northern hemisphere.

Our results here provide additional possible players in picture above. Solar activity can be linked to stratosphere (see for example, Climate variability and sunspot activity: Analysis of solar influence on climate. Indrani Roy, Editor, Springer, ISBN 978-3-319-77106-9). Solar activity impacts the QBO and thus the stratosphere, which together with ENSO are implicated in this 3-D loop. Our results provide further new insights into those dynamical mechanisms and how the complex interactions among the base driving factors and their harmonics may cause the peak-periods in climate modes and thus affect climate variability.

We will make these points in the revised manuscript.

---

## Author Comment (AC3) · 20 Feb 2020

We sincerely appreciate the reviewer's helpful and insightful comments. We tried our best to revise our manuscript accordingly. This referee had common comments with referee #1. Therefore, to avoid simply duplicating responses, we kindly ask the referee to consult out reply to referee #1.

As we reply to the first referee, varying the embedding dimension provides a sensitivity analysis to show the robustness of our results. We also provided results from an ideal model to demonstrate the effectiveness of the technique of combining SFA with wavelet analysis. The significant peak-periods of SFA signals can reflect the true driving force factors very well. However, we note that other methods may be developed to quantify

the uncertainties of the peak periods in future studies. As for the physical/dynamical mechanisms, we have expanded our discussion in the end of our manuscript.

This is a relatively new area of research and more insights we hope will be forthcoming in future work.

---

## Author Response (AR1)

Dear Dr. Kravitz,

We have submitted our revised manuscript. In the revised paper, we have included parts to address the comments and our replies to the reviewers 1, and 2. We have thus, included parts that address the issues of the robustness of the results, and of the physical/dynamical mechanisms that may be involved (pages 12 and 15-16, new figure 5, and note 1 in supplementary material). Also, in supplementary material note 2 we addressed your comment about "the logistic example being a representative result". In supplementary note 3 we have briefly addressed Prof. Sonechkin's interactive comment.

We hope that you will find our replies and revision more than adequate and that the paper will be accepted for publication.

Sincerely,

Xinnong Pan

Geli Wang

Anastasios Tsonis

[revised manuscript text omitted]

shows the normalized series of these climate indices. These indices and their corresponding climate modes are described briefly as follows.

**2.1 ENSO**

ENSO is well recognized as a natural ocean-atmosphere coupled mode in the tropical Pacific (Deser et al.,

2010), affecting the global climate (Newman et al., 2003). El Niño (La Niña) refers to warming (cooling)

phase of the tropical Pacific Ocean occurring every 2–7 yr. Meanwhile, the anomalous warming or cooling conditions are linked to a large-scale east-west seesaw air pressure pattern, referred to Southern Oscillation (Capotondi et al., 2015). El Niño and Southern Oscillation are two manifestations of ENSO phenomenon (Bjerknes, 1969). In this study, ENSO is represented by both the Niño 3.4 index and the Southern Oscillation

Indices (SOI). The Niño 3.4 index (1870/01–2018/12, hereafter referred to as NINO) is defined as the SST

anomalies in the Niño 3.4 region (5ºN–5ºS; 170–120ºW) based on the HadISST1 dataset (Rayner et al., 2003).

The SOI index (1866/01–2017/12) is calculated from the observed standardized sea level pressure (SLP)

differences between the islands of Tahiti and Darwin, Australia (Ropelewski et al., 1987).

**2.2 PDO**

PDO is the dominant pattern of decadal variability of North Pacific SST, which has been widely-studied across different disciplines (Newman et al., 2016). Previous study shows that the changing phase of PDO affects the anomalies of atmospheric circulation around North Pacific Ocean basin, and even the South Hemisphere (Mantua and Hare, 2002). The characteristic period of PDO is 50–60 yr and a warm or cold phase of PDO can typically persist for about 20–30 yr. If PDO is in its positive phase, the North Pacific Ocean turns colder and

Middle East Pacific Ocean turns warmer, otherwise it is in negative phase. In this study, PDO is defined by the leading principal component of monthly SST anomalies in the Pacific basin (poleward of 20ºN) during

1900–2017 (Mantua et al., 1997).

**2.3 AMO**

AMO is a dominant signal of climate variability in the field of North Atlantic SST, which has a statistically significant spectral peak in the 50-70 yr band (Schlesinger et al., 1994; Sun et al., 2015). Related studies suggested that AMO is an inner variability of climate system, modulating hemispheric climate change (Zhang, 2007; Knight et al., 2006). The slow variation of the Atlantic meridional overturning circulation (AMOC) plays a dominant role in the Atlantic multidecadal variability of SST (Zhang, 2017; Delworth et al., 2000; Garuba et al., 2018). The AMO is defined by the detrended area-weighted average SST over the North Atlantic (from 0° to 70°N) during 1856–2018 based on the Kaplan SST dataset (Enfield et al., 2001). Both unsmoothed and smoothed AMO indexes are available. The high-frequency variability of the smoothed AMO index has been removed by a common 121-month filter. We choose to use the unsmoothed AMO index in this study.

**2.4 NAO**

The NAO is active in the North Atlantic region that is characterized by a large-scale seesaw in atmospheric mass between the subtropical high and the polar low (Li et al., 2003). It manifests as climate fluctuations at multiple timescales ranging from inter-annual to multi-decadal variabilities (Jones et al., 1997; Li et al., 2013), affecting the climate within and around North Atlantic Ocean basin, and even the entire Northern Hemisphere (Wallace and Gutzler, 1981; Hurrell,1995; Li et al., 2013; Delworth et al., 2016; Jajcay et al., 2016). Although the climatic effect of NAO is most pronounced in winter, it is the dominant mode of atmospheric circulation in the North Atlantic sector throughout the whole year. Previous study suggested that NAO drives the North Atlantic SST anomalies at a timescale less than 10 yr (Delworth et al., 2017). NAO index is typically defined as a meridional dipole mode (which has been lately suggested of being a three-pole pattern (Tsonis et al., 2008)) in atmospheric pressure with two centers of action in Iceland and Azores during 1825–2017. For comparison, we also examine another observationally-based monthly NAO index for the period 1850–2015

(hereafter referred to as NAOI), which is defined by the difference in the normalized sea level pressure (SLP)

that is zonally-averaged over the North Atlantic sector from 80ºW to 30ºE between 35ºN and 65ºN (Li et al.,

2003; http://ljp.gcess.cn/dct/page/65610). The NAOI is calculated based on the HadSLP dataset with the reference period of 1961–1990.

**3 Methods**

**3.1 Slow Feature Analysis (SFA)**

Based on time-embedding theorems, one-dimensional time series can turn into a multidimensional system.

For this multidimensional input system, the SFA acts as a nonlinear method that uses a nonlinear expansion to map the input signal into a feature space and solves a linear problem (Blaschke et al., 2006). The objective of SFA is to find instantaneous scalar input-output functions that generate output signals that vary as slowly as possible but still carry significant information. To ensure this, we require the output signals to be uncorrelated and have unit variance (Franzius et al., 2011).

Consider a time series $\{x(t)\}_{t=t_1,\ldots,t_n}$, where $t$ denotes time and $n$ indicates the length of the time series. First, we embed $\{x(t)\}$ into an $m$-dimensional state space:

$$\mathbf{X}(t) = \{x_1(t), x_2(t), \ldots, x_m(t)\}_{t=t_1,\ldots,t_N}$$

where $N = n - m + 1$. Then nonlinear expansions (usually second-order polynomials) are used to generate a k- dimensional function state space:

$$\mathbf{H}(t) = \{x_1(t), \ldots, x_m(t), x_1^2(t), \ldots, x_1(t)x_m(t), \ldots, x_{m-1}^2(t), \ldots, x_m^2(t)\}_{t=t_1,\ldots,t_N},$$

which can also be written as $\mathbf{H}(t) = \{h_1(t), h_2(t), \ldots, h_k(t)\}_{t=t_1,\ldots,t_N}$, where

$$k = m + m(m + 1)/2.$$

Then, the expanded signal $\mathbf{H}(t)$ is normalized so that it satisfies the constraints of zero mean and unit variance. This process is referred to as whitening or sphering. Thus, we have

$$\mathbf{H}'(t) = \{h_1'(t), h_2'(t), \dots, h_k'(t)\}_{t=t_1,\dots,t_N}, \text{ where}$$

$$\overline{h_j'} = 0 \text{ (zero mean)},$$

$$h_j' h_j'^T = 1 \text{ (unit variance)},$$

$$h_j'(t) = \left[h_j(t) - \overline{h_j}\right]/S, \text{ and } S = \frac{1}{k}\sqrt{\sum_{j=1}^{k}(h_j(t) - \overline{h})^2}.$$

Using Schmidz algorithm, $\mathbf{H}'(t)$ is orthogonized into:

$$\mathbf{Z}(t) = \{z_1(t), z_2(t), \dots, z_k(t)\}_{t=t_1,\dots,t_N}.$$

Thus, each output signal can be expressed as the following linear combination:

$$y(t) = a_1 z_1(t) + a_2 z_2(t) + \dots + a_k z_k(t),$$

$(a_1, a_2, \dots, a_k)$ is a set of weighting coefficients.

Note that the output signals are orthogonal and nontrivial:

$$z_i(t) \cdot z_j(t) = 0, \ \overline{z_i}(t) = \overline{z_j}(t) = 0, \ z_j(t) \cdot z_j^T(t) = 1,$$

Subsequently, we perform the 1$^{st}$ order differencing on $\mathbf{Z}(t)$ to obtain the derivative function space:

$$\dot{z}_j(t_i) = z_j(t_{i+1}) - z_j(t_i)$$

[revised manuscript text omitted]

**5 Conclusions and discussions**

In this study, we identify four independent base peak-periods: $T_q$ (2.32 yr), $T_{e1}$ (3.90 yr), $T_{e2}$ (6.55 yr) and $T_s$ (11.02 yr). We infer that these base peak-periods are essentially associated with the QBO cycle, two intrinsic ENSO cycles and the solar cycle, respectively. Other detected significant peak-periods can be represented by the integral multiples of these four base periods. This implies that the QBO, ENSO and solar activities could be three key periodic driving factors in global climate variability. These results provide possible clues for the intricate relationships between driving forces and their harmonics in the variability of major climate modes as well as the coupling ways among them. The finding of the interconnections of major climate modes indicates that using statistical models to predict the decadal-to-multidecadal climate variability is promising in the future. It should be noted that uncertainties still exist in the multidecadal variability of ENSO and QBO. The relatively long peak-periods (e.g. 52.42 yr, 62.33 yr, 74.13 yr and 88.15 yr) detected by SFA may be resulted from the effect of continuous wavelet transform.

Recent studies on complex climate networks provided new insights into how the collective behavior of major climate modes affects global temperature variations (Tsonis et al., 2007; Tsonis 2018). By considering a network of major climate modes (more or less the same set as here and the theory of synchronized chaos, these previous studies found that the network may synchronize temporally. During synchronization, the increased coupling strength among the climate modes may lead to the destruction of the synchronized state that leads to changes in the trends of global temperature and the amplitudes of ENSO variability on decadal-to-multidecadal timescales. These studies proposed a dynamical mechanism and its related physical causes for the observed climate shifts. The idea that the interaction between major climate modes play a significant role in climate variability has in the last decade or so found many applications.

Solid dynamical arguments and past work offer a concrete picture of how the physics may play out (Wang et al., 2009). NAO with its huge mass re-arrangement in north Atlantic affects the strength of the westerly flow across mid-latitudes. At the same time through its ''twin'', the arctic Oscillation (AO), it impacts sea level pressure patterns in the northern Pacific. This process is part of the so-called intrinsic mid-latitude northern hemisphere variability. Then this intrinsic variability through the seasonal ''footprinting'' mechanism couples with equatorial wind stress anomalies, thereby acting as a stochastic forcing of ENSO. Subsequently, ENSO with its effects on PNA can through vertical propagation of Rossby waves influence the lower stratosphere and in turn the stratosphere influence NAO through downward progression of Rossby waves. These results coupled with our results suggest the following 3-D super-loop NAO → PDO → ENSO → PNA → stratosphere → NAO, which may capture the essence of low-frequency variability in the northern hemisphere (**Fig. 5**).

While still more work is needed on the physical/dynamical links between major climate modes and their interactions, our results here provide additional possible players in picture above. Solar activity can be linked to stratosphere (see (Roy, 2018) for example). Solar activity impacts the QBO and thus the stratosphere, which together with ENSO are implicated in this 3-D loop. Our results provide further new insights into those dynamical mechanisms and how the complex interactions among the base driving factors and their harmonics may cause the peak-periods in climate modes and thus affect climate variability.

***Code/Data availability***. All data needed to evaluate the conclusions in the paper are present in the paper. Additional data and codes related to this paper may be requested from the corresponding author.

***Author contribution.*** Xinnong Pan and Geli Wang designed this study. All of the authors contributed to the preparation and writing of the manuscript.

***Competing interests.*** The authors declare no competing interest.

***Acknowledgements.*** This research was supported by the National Key R&D Program of China (2017YFC1501804), the National Natural Science Foundation of China (91737102 and 41575058).

[revised manuscript text omitted]